# RETHINKING THE EFFECT OF DATA AUGMENTATION IN ADVERSARIAL CONTRASTIVE LEARNING

**Rundong Luo**[1]* **Yifei Wang**[2]* **Yisen Wang**[3,4]†

[1]School of EECS, Peking University
[2]School of Mathematical Sciences, Peking University
[3]National Key Lab. of General Artificial Intelligence,
 School of Intelligence Science and Technology, Peking University
[4]Institute for Artificial Intelligence, Peking University

## ABSTRACT

Recent works have shown that self-supervised learning can achieve remarkable robustness when integrated with adversarial training (AT). However, the robustness gap between supervised AT (sup-AT) and self-supervised AT (self-AT) remains significant. Motivated by this observation, we revisit existing self-AT methods and discover an inherent dilemma that affects self-AT robustness: either strong or weak data augmentations are harmful to self-AT, and a medium strength is insufficient to bridge the gap. To resolve this dilemma, we propose a simple remedy named DYNACL (Dynamic Adversarial Contrastive Learning). In particular, we propose an augmentation schedule that gradually anneals from a strong augmentation to a weak one to benefit from both extreme cases. Besides, we adopt a fast post-processing stage for adapting it to downstream tasks. Through extensive experiments, we show that DYNACL can improve state-of-the-art self-AT robustness by 8.84% under Auto-Attack on the CIFAR-10 dataset, and can even outperform vanilla supervised adversarial training for the first time. Our code is available at `https://github.com/PKU-ML/DYNACL`.

## 1 INTRODUCTION

Learning low-dimensional representations of inputs without supervision is one of the ultimate goals of machine learning. As a promising approach, self-supervised learning is rapidly closing the performance gap with respect to supervised learning (He et al., 2016; Chen et al., 2020b) in downstream tasks. However, for whatever supervised and self-supervised learning models, adversarial vulnerability remains a widely-concerned security issue, *i.e.,* natural inputs injected by small and human imperceptible adversarial perturbations can fool the deep neural networks (DNNs) into making wrong predictions (Goodfellow et al., 2014).

In supervised learning, the most effective approach to enhance adversarial robustness is adversarial training (**sup-AT**) that learns DNNs with adversarial examples (Madry et al., 2017; Wang et al., 2019; Zhang et al., 2019; Wang et al., 2020; Wang & Wang, 2022). However, sup-AT requires groundtruth labels to craft adversarial examples. In self-supervised learning, recent works including RoCL (Kim et al., 2020), ACL (Jiang et al., 2020), and AdvCL (Fan et al., 2021) explored some adversarial training counterparts (**self-AT**). However, despite obtaining a certain degree of robustness, there is still a very large performance gap between sup-AT and self-AT methods. As shown in Figure 1(a), sup-AT obtains 46.2% robust accuracy while state-of-the-art self-AT method only gets 37.6% on CIFAR-10, and the gap is $> 8\%$. As a reference, in standard training (ST) using clean examples, the gap in classification accuracy between sup-ST and self-ST is much smaller (lower than $1\%$ on CIFAR-10, see da Costa et al. (2022)). This phenomenon leads to the following question:

*What is the key factor that prevents self-AT from obtaining comparable robustness to sup-AT?*

To answer this question, we need to examine the real difference between sup-AT and self-AT. As they share the same minimax training scheme, the difference mainly lies in the learning objective.

---

*Equal Contribution.
†Corresponding Author: Yisen Wang (yisen.wang@pku.edu.cn).

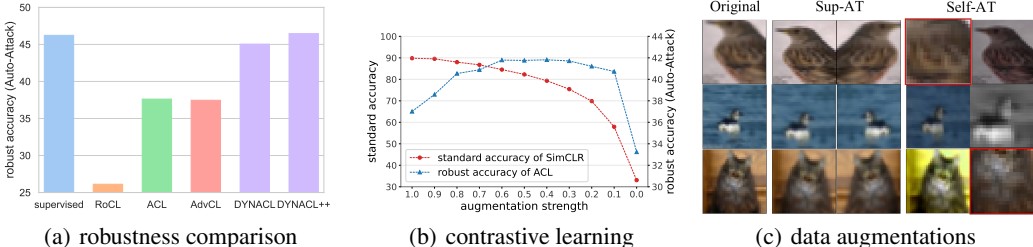

|(a) robustness comparison|(b) contrastive learning|(c) data augmentations|

Figure 1: Experiments on CIFAR-10: (a) Comparison of supervised AT (vanilla PGD-AT (Madry et al., 2017)) and five self-supervised AT methods: RoCL, ACL, AdvCL, our DYNACL and DYNACL++ with ResNet-18 backbone. (b) Performance of standard contrastive learning (Chen et al., 2020a) and adversarial contrastive learning (Jiang et al., 2020) using different augmentation strengths. (c) Illustrative examples of data augmentations adopted by sup-AT and self-AT. We can see that self-AT adopts much more aggressive augmentation than sup-AT.

Different from sup-AT relying on labels, self-AT methods (RoCL, ACL, and AdvCL) mostly adopt contrastive learning objectives, which instead rely on matching representations under (strong) data augmentations to learn meaningful features. As noted by recent theoretical understandings (HaoChen et al., 2021; Wang et al., 2022), the core mechanism of contrastive learning is that it leverages strong augmentations to create support overlap between intra-class samples, and as a result, the alignment between augmented positive samples could implicitly cluster intra-class samples together. Weak augmentations, instead, are not capable of generating enough overlap, and thus lead to severe performance degradation (Figure 1(b) red line). In the meanwhile, we also find that strong data augmentations can be very harmful for adversarial robustness (Figure 1(b) blue line). This reveals a critical dilemma of self-AT, that (default) strong augmentations, although useful for standard accuracy, also severely hurt adversarial robustness. To understand why, we draw some augmented samples and observe that self-AT augmentations induce significant semantic shifts to original images (Figure 1(c)). This suggests that strong augmentations create a very large distribution shift between training and test data, and the local $\ell_p$ robustness on training examples becomes less transferable to test data. Notably, some augmented samples from different classes can even become inseparable (the bird and the cat samples in red squares), and thus largely distort the decision boundary. The necessity and harmfulness of data augmentations form a fundamental dilemma in self-AT.

Therefore, as data augmentations play an essential role in self-AT, we need to strike a balance between utilizing strong augmentations for representation learning and avoiding large image distortions for good robustness. Although it seems impossible under the current static augmentation strategy, we notice that we can alleviate this dilemma by adopting a *dynamic* augmentation schedule. In particular, we could learn good representations with aggressive augmentations at the beginning stage, and gradually transfer the training robustness to the test data by adopting milder and milder augmentations at later stages. We name this method Dynamic Adversarial Contrastive Learning (DYNACL). In this way, DYNACL could benefit from both sides, gaining both representation power and robustness aligned with test distribution. Built upon DYNACL, we further design a fast post-processing stage for bridging the difference between the pretraining and downstream tasks, dubbed DYNACL++. As a preview of the results, Figure 1(a) shows that the proposed DYNACL and DYNACL++ bring a significant improvement over state-of-the-art self-AT methods, and achieve comparable, or even superior robustness, to vanilla sup-AT. Our main contributions are:

- We reveal the reason behind the robustness gap between self-AT and sup-AT, *i.e.,* the widely adopted aggressive data augmentations in self-supervised learning may bring the issues of training-test distribution shift and class inseparability.

- We propose a dynamic augmentation strategy along the training process to balance the need for strong augmentations for representation and mild augmentations for robustness, called Dynamic Adversarial Contrastive Learning (DYNACL) with its variant DYNACL++.

- Experiments show that our proposed methods improve both clean accuracy and robustness over existing self-AT methods by a large margin. Notably, DYNACL++ improves the robustness of ACL (Jiang et al., 2020) from 37.62% to 46.46% on CIFAR-10, which is even

slightly better than vanilla supervised AT (Madry et al., 2017). Meanwhile, it is also more computationally efficient as it requires less training time than ACL.

## 2 BACKGROUND AND RELATED WORK

**Sup-AT.** Given a labeled training dataset $\mathcal{D}_l = \{(\bar{x}, y)|\bar{x} \in \mathbb{R}^n, y \in [K]\}$, to be resistant to $\ell_p$-bounded adversarial attack, supervised adversarial training (sup-AT) generally adopts the following min-max framework (Madry et al., 2017):

$$\mathcal{L}_{\text{sup}-\text{AT}}(f) = \mathbb{E}_{\bar{x},y} \max_{\delta_{\bar{x}} \in \Delta} \ell_{\text{CE}}(f(\bar{x} + \delta_{\bar{x}}), y), \quad \ell_{\text{CE}}(f(\cdot), y) = -\log \frac{\exp(f(\cdot)[y])}{\sum_{k=1}^{K} \exp(f(\cdot)[k])}. \quad (1)$$

Here the classifier $f : \mathbb{R}^n \to \mathbb{R}^K$ is learned on the adversarially perturbed data $(\bar{x} + \delta_{\bar{x}}, y)$, and $\Delta = \{\delta : \|\delta\|_p \leq \varepsilon\}$ denotes the feasible set for adversrial perturbation.

**Contrastive Learning.** For each $\bar{x} \in \mathcal{X}$, we draw two positive samples $x, x^+$ from the augmentation distribution $A(\cdot|\bar{x})$, and draw $M$ independent negative samples $\{x_m^-\}_{m=1}^M$ from the marginal distribution $A(\cdot) = \mathbb{E}_{\bar{x}} A(\cdot|\bar{x})$. Then, we train an encoder $g : \mathcal{X} \to \mathcal{Z}$ by the widely adopted InfoNCE loss using the augmented data pair $(x, x^+, \{x_m^-\})$ (Oord et al., 2018):

$$\min_g \mathcal{L}_{\text{NCE}}(g) = \mathbb{E}_{x,x^+,\{x_m^-\}} \ell_{\text{NCE}}(x, x^+, \{x_m^-\}; g),$$

$$\ell_{\text{NCE}}(x, x^+, \{x_m^-\}; g) = -\log \frac{\exp(\text{sim}(g(x), g(x^+))/\tau)}{\sum_m \exp(\text{sim}(g(x), g(x_m^-))/\tau)}. \quad (2)$$

Here $\text{sim}(\cdot, \cdot)$ is the cosine similarity between two vectors, and $\tau$ is a temperature hyperparameter. Surged from InfoNCE (Oord et al., 2018), contrastive learning undergoes a rapid growth (Tian et al., 2019; Chen et al., 2020a; He et al., 2020; Wang et al., 2021) and has demonstrated state-of-the-art performance on self-supervised tasks (Chen et al., 2020b; Kong et al., 2020). Nevertheless, several works also point out contrastive learning is still vulnerable to adversarial attack when we transfer the learned features to the downstream classification (Ho & Nvasconcelos, 2020; Kim et al., 2020).

**Self-AT.** To enhance the robustness of contrastive learning, adversarial training was similarly adapted to self-supervised settings (self-AT). Since there are no available labels, adversarial examples are generated by maximizing the contrastive loss (Eq. 2) *w.r.t.* all input samples,

$$\ell_{\text{AdvNCE}}(x, x^+, \{x_m^-\}; g) = \max_{\delta, \delta^+, \{\delta_m^-\} \in \Delta} \ell_{\text{NCE}}(x + \delta, x^+ + \delta^+, \{x_m^- + \delta_m^-\}; g), \quad (3)$$

Several previous works, including ACL (Jiang et al., 2020), RoCL (Kim et al., 2020), and CLAE (Ho & Nvasconcelos, 2020), adopt Eq. 3 to perform self-AT. In addition, ACL (Jiang et al., 2020) further incorporates the dual-BN technique (Xie et al., 2020) for better performance. Specifically, given a backbone model like ResNet, they duplicate its BN modules and regard the encoder $g$ as two different branches (all parameters are shared except for BN): the clean branch $g_c$ and the adversarial branch $g_a$ are trained by clean examples and adversarial examples respectively:

$$\ell_{\text{ACL}}(x, x^+, \{x_m^-\}; g) = \ell_{\text{NCE}}(x, x^+, \{x_m^-\}; g_c) + \ell_{\text{AdvNCE}}(x, x^+, \{x_m^-\}; g_a). \quad (4)$$

After training, they use the adversarial branch $g_a$ as the robust encoder. They empirically show that ACL obtains better robustness than the single branch version. Built upon ACL, AdvCL (Fan et al., 2021) further leverages an additional ImageNet-pretrained model to generate pseudo-labels for sup-AT via clustering. Very recently, DeACL (Zhang et al., 2022) proposes a two-stage method, that is to distil a standard pretrained encoder via adversarial training. In this paper, following the vein of RoCL, ACL, and AdvCL, we study how to perform adversarial contrastive learning from scratch.

## 3 RETHINKING THE EFFECT OF DATA AUGMENTATIONS IN SELF-AT

Different from the case of supervised learning, data augmentation is an *indispensable* ingredient in contrastive self-supervised learning. Existing self-AT methods inherit the aggressive augmentation strategy adopted in standard contrastive learning (*e.g.,* SimCLR (Chen et al., 2020a)), *i.e.,* a composition of augmentations like random resized crop, color jitter, and grayscale. However, this strategy is *not* necessarily optimal for adversarial training. In this section, we provide an in-depth investigation of the effect of this default data augmentation strategy on self-supervised adversarial training (self-AT).

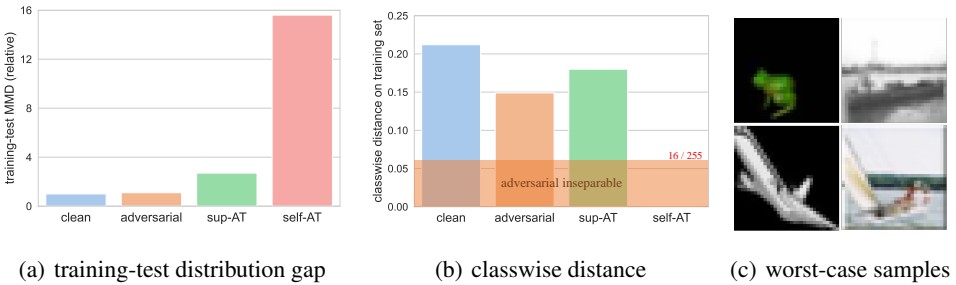

(a) training-test distribution gap    (b) classwise distance    (c) worst-case samples

Figure 2: (a & b) Comparison of different augmentation strategies on CIFAR-10, where "clean" denotes no augmentation; "adversarial" denotes adversarial examples generated by PGD attack; "sup-AT" and "self-AT" denote the corresponding default data augmentations. (c) Training examples from different classes of CIFAR-10 that could yield pure black or pure white augmented views.

### 3.1 PROBLEMS OF AGGRESSIVE DATA AUGMENTATIONS IN SELF-AT

As shown in Figure 1(c), the default augmentations in self-AT are much more aggressive than that of sup-AT, which leads to two obvious drawbacks: large training-test distribution gap and class inseparability, as we quantitatively characterize below. For comparison, we evaluate 4 kinds of data augmentation methods on CIFAR-10: 1) clean (no augmentation), 2) pre-generated adversarial augmentation by a pretrained classifier, 3) the default augmentations (padding-crop and horizontal flip) adopted by sup-AT (Madry et al., 2017), and 4) the default augmentations (random resized crop, color jitter, grayscale, and horizontal flip) adopted by self-AT (Jiang et al., 2020; Kim et al., 2020).

**Large Training-test Distribution Gap.** Although training and test samples are supposed to follow the same distribution, random augmentations are often adopted in the training stage, which creates a training-test distribution gap. To study it quantitatively in practice, we measure this distribution gap with Maximum Mean Discrepancy (MMD) (Gretton et al., 2012) calculated between the augmented training set and the raw test set of CIFAR-10 (details in Appendix B.1).[1] As shown in Figure 2(a), "adversarial" and "sup-AT" augmentations only produce a small distribution gap, while the aggressive "self-AT" augmentations yield a much larger gap (6 times) than "sup-AT". Therefore, the self-AT training data could be very distinct from test data, and the robustness obtained *w.r.t.* augmented training data could be severely degraded when transferred to the test data. Due to the limit of space, a theoretical justification of this point is included in Appendix E.

**Class Inseparability.** Yang et al. (2020) found that under supervised augmentations and adversarial perturbations with $\varepsilon = 8/255$, the minimal $\ell_\infty$ distance between training samples from different classes, namely the *classwise distance*, is larger than $2 \cdot \varepsilon = 16/255$. In other words, CIFAR-10 data are *adversarially separable* in sup-AT. However, we notice that this *no longer* holds in self-AT. As shown in Figure 2(b), self-AT augmentations have a much smaller classwise distance that is close to zero. One possible cause is those CIFAR-10 samples with complete black or white regions, which could result in identical augmentations after aggressive random cropping (Figure 2(c)). As a result of class inseparability, self-AT will mix adversarial examples from different classes together, and these noisy examples will degrade the robustness of the learned decision boundary.

### 3.2 THE DILEMMA OF SELF-AT UNDER STATIC AUGMENTATION STRENGTH

The discussion above shows that aggressive augmentations are harmful to self-AT, which motivates us to quantitatively study the effect of different augmentation strengths on self-AT.

**Quantifying Augmentation Strength.** In order to give a quantitative measure of the augmentation strength, we design a set of data augmentations $\mathcal{T}(s)$ that vary according to a hyperparameter $s \in [0, 1]$: 1) when $s = 0$, it is mild and contains only horizontal flip; 2) when $s = 1$, it is the aggressive augmentations of self-AT; and 3) when $s \in (0, 1)$, we linearly interpolate the augmentation hyperparameters to create a middle degree of augmentation. For example, in random resized cropping,

---

[1]As adversarial perturbations are defined in the input space, we measure the distance in the input space to evaluate adversarial vulnerability. We include latent-space results in Appendix B.2, which show similar trends.

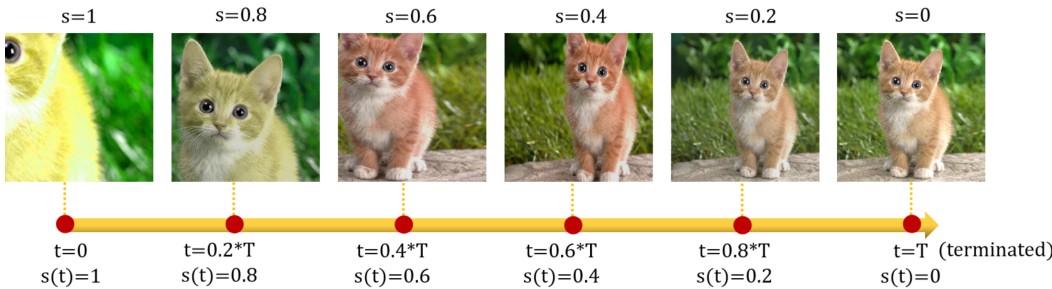

Figure 3: Illustration of augmenting the same image of a cat with different augmentation strengths $s \in [0, 1]$, where a smaller $s$ (left to right) indicates milder image distortion.

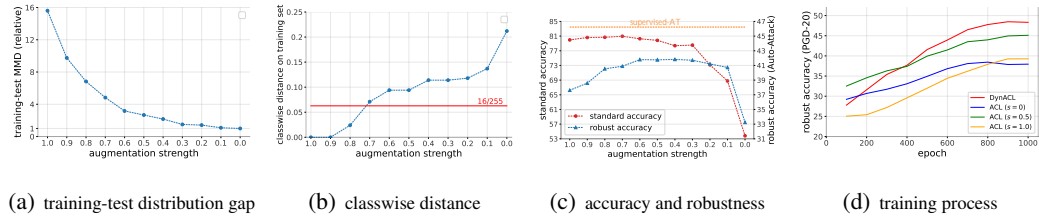

(a) training-test distribution gap     (b) classwise distance     (c) accuracy and robustness     (d) training process

Figure 4: Quantitative results on CIFAR-10 of self-AT (baseline method is ACL (Jiang et al., 2020)) with different augmentation strengths.

a larger $s$ will crop to a smaller region in an image, which amounts to a larger training-test distribution gap, as shown in Figure 3. See a detailed configuration of $\mathcal{T}(s)$ in Appendix A.1.

**Static Augmentation Cannot Fully Resolve the Dilemma of Self-AT.** The discussion in Section 3.1 reveals that strong augmentations are harmful to adversarial training while being critical for representation learning (Figure 1(b)). The augmentation strength thus becomes a critical dilemma of self-AT. Here, we study whether we could resolve this issue by tuning the augmentation strength $s$ for an optimal tradeoff. As shown in Figures 4(a) and 4(b), decreasing augmentation strength $s$ (left to right) indeed brings smaller training-test distribution gap and eliminates class inseparability, which benefits adversarial robustness (blue line in Figure 4(c)) and hurts standard accuracy at the same time (the red line in Figure 4(c)). Roughly, $s = 0.5$ seems a good trade-off between accuracy and robustness, while its 41.79% robustness is still much lower than the 46.23% robustness of sup-AT (orange line in Figure 4(c)). This suggests that the dilemma of augmentation strength cannot be fully resolved by tuning a static augmentation strength. The training process in Figure 4(d) presents that strong or weak augmentations have their own defects while medium augmentations only bring a limited gain (green line). In the next section, we will introduce a new dynamic augmentation strategy to resolve this dilemma (a preview of performance is shown as the red line in Figure 4(d)).

## 4   DYNAMIC ADVERSARIAL CONTRASTIVE LEARNING (DYNACL)

In this section, we propose a simple but effective method **DYNACL** to resolve the dilemma of self-AT augmentations. In Section 4.1, we introduce our dynamic pretraining method for self-AT, named DYNACL. In Section 4.2, we further propose an improved version of DYNACL with a fast post-processing stage to mitigate the gap between pretraining and downstream tasks, named DYNACL++. Figure 5 in Appendix A.3 demonstrates the framework of the proposed DYNACL.

### 4.1   DYNACL: DYNAMIC ADVERSARIAL CONTRASTIVE LEARNING

In view of the dilemma of static augmentation strength (Section 3.2), we propose a new *dynamic* training recipe for self-AT, named Dynamic Adversarial Contrastive Learning (**DYNACL**), which consists of a dynamic schedule for data augmentation and a dynamic objective for model training.

**Dynamic Augmentation.** Different from prior works that all adopt a static augmentation strength, we propose to treat the augmentation strength $s$ as a dynamic parameter $s(t)$ that varies along different epochs $t$. In this way, at earlier training stages, we can enforce self-AT to learn meaningful representations by adopting aggressive augmentations with a large augmentation strength $s(t)$. Afterwards, we could gradually mitigate the training-test distribution gap and class inseparability by annealing down the augmentation strength $s(t)$ (see Figures 4(a) and 4(b)). With this dynamic augmentation schedule, we could combine the best of both worlds: on the one hand, we can benefit from the representation ability of contrastive learning at the beginning stage; on the other hand, we can achieve better adversarial robustness by closing the distribution gap at the later stage.

Following the analysis above, we design the following piecewise decay augmentation schedule:

$$s(t) = 1 - \lfloor \frac{t}{K} \rfloor \cdot \frac{K}{T}, \quad t = 0, \ldots, T - 1, \tag{5}$$

where $T$ is the total number of training epochs, $K$ is the decay period, and $\lfloor \cdot \rfloor$ is the floor operation. As illustrated in Figure 3, this schedule $s(t)$ starts from strong augmentations with $s = 1$ at the start epoch, and gradually decays to a weak augmentation with $s = 0$ by reducing $K/T$ every $K$ epochs.

**Dynamic ACL Loss.** Following the same idea above, we can also design a dynamic version of the ACL loss in Eq. 4. Instead of adopting an equal average of the clean and adversarial contrastive losses, we can learn meaningful representations in the beginning stage with the clean contrastive loss $\ell_{\text{NCE}}$ (Eq. 2). Later, we can gradually turn our focus to adversarial robustness by down-weighting the clean contrastive loss and up-weighting the adversarial loss $\ell_{\text{AdvNCE}}$ (Eq. 3). Following this principle, we propose a dynamic ACL loss with a dynamic reweighting function $w(t)$ for the $t$-th epoch:

$$\mathcal{L}_{\text{DYNACL}}(g; t) = (1 - w(t))\mathcal{L}_{\text{NCE}}(g_c; s(t)) + (1 + w(t))\mathcal{L}_{\text{AdvNCE}}(g_a; s(t)). \tag{6}$$

For simplicity, we assign the dynamic reweighting function $w(t)$ as a function of the dynamic augmentation schedule $s(t)$:

$$w(t) = \lambda(1 - s(t)), \quad t = 0, \ldots, T - 1, \tag{7}$$

where $\lambda \in [0, 1]$ denotes the reweighting rate. In this way, when the training starts ($t = 0$), we adopt an equal combination of the clean and adversarial losses as in ACL. As training continues, the augmentation strength $s(t)$ anneals down, and the reweighting coefficient $w(t)$ becomes larger. Accordingly, the clean loss $\ell_{\text{NCE}}$ will have a smaller weight while the adversarial loss $\ell_{\text{AdvNCE}}$ will have a larger weight. At last ($t = T$), the weight of clean loss is decreased to $1 - \lambda$, and the weight of adversarial loss is increased to $1 + \lambda$. A larger reweighting rate $\lambda$ indicates more aggressive reweighting. Our dynamic ACL loss degenerates to vanilla ACL loss with $\lambda = 0$ as a special case.

## 4.2 DYNACL++: DYNACL with Fast Post-processing

The main dynamic pretraining phase proposed above can help mitigate the training-test data distribution gap. Nevertheless, there remains a training-test task discrepancy lying between the contrastive pretraining and downstream classification tasks: the former is an *instance-level* discriminative task, while the latter is a *class-level* discriminative task. Therefore, the pretrained instance-wise robustness might not transfer perfectly to the classwise robustness to be evaluated at test time.

To achieve this, we propose a simple post-processing phase with Pseudo Adversarial Training (PAT) to mitigate this "task gap". Given a pretrained encoder with a clean branch $g_c$ and an adversarial branch $g_a$, we simply generate pseudo labels $\{\hat{y}_i\}$ by k-means clustering using the clean branch $g_c$, which natural accuracy is relatively high because it is fed with natural samples during training. Inspired by the Linear Probing then full FineTuning (LP-FT) strategy (Kumar et al., 2022), we propose to learn a classifier with Linear Probing then *Adversarially* full FineTuning (LP-AFT) for better feature robustness: we first use standard training to train a linear head $h : \mathbb{R}^m \to \mathbb{R}^k$ ($k$ is the number of pseudo-classes) with fixed adversarial branch $g_a$, and then adversarially finetune the entire classifier $h \circ g_a$ on generated pseudo pairs $(x, \hat{y})$ using the TRADES loss (Zhang et al., 2019). In practice, we only perform PAT for a few epochs (*e.g.,* 25), which is much shorter than the pretraining stage with typically 1000 epochs. Thus, we regard this as a fast post-processing stage of the learned features to match the downstream classification task and name the improved version DYNACL++.

**Discussions on Previous Works.** Our DYNACL is built upon the dual-stream framework of ACL (Jiang et al., 2020). Compared to ACL with static augmentation strength and learning objective, DYNACL adopts a dynamic augmentation schedule together with a dynamic learning objective.

Further, DYNACL++ recycles the clean branch of ACL to further bridge the task gap with a fast post-processing stage. Also built upon ACL, AdvCL (Fan et al., 2021) incorporates a third view for high-frequency components, and incorporates the clustering process into *the entire pretraining phase*. Thus, AdvCL requires much more training time than ACL, while our dynamic training brings little computation overhead. Additionally, AdvCL generates pseudo labels by an ImageNet-pretrained model before pretraining, while DYNACL++ generates pseudo labels by the pretrained model itself. Moreover, even with ImageNet labels, AdvCL still performs inferior to our DYNACL and DYNACL++ (Section 5.1). Lastly, we devise a Linear-Probing-Adversarial-Full-Finetuning (LP-AFT) strategy to exploit pseudo labels, while AdvCL simply use them as pseudo supervision.

## 5 EXPERIMENTS

In this section, we evaluate DYNACL and DYNACL++ under the benchmark datasets: CIFAR-10, CIFAR-100 (Krizhevsky, 2009), and STL-10 (Coates et al., 2011), compared with baseline methods: RoCL (Kim et al., 2020), ACL (Jiang et al., 2020), and AdvCL (Fan et al., 2021). We provide additional experimental details in Appendix C.

**Pretraining.** We adopt ResNet-18 (He et al., 2016) as the encoder following existing self-AT methods (Kim et al., 2020; Jiang et al., 2020; Fan et al., 2021). We set the decay period $K = 50$, and reweighting rate $\lambda = 2/3$. As for other parts, we mainly follow the training recipe of ACL (Jiang et al., 2020), while we additionally borrow two training techniques from modern contrastive learning variants: the stop gradient operation and the momentum encoder. Both are proposed by BYOL (Grill et al., 2020) and later incorporated in MoCo-v3 (Chen et al., 2021) and many other methods. An ablation study on these two components can be found in Appendix D.3, where we show that the improvement of DYNACL mainly comes from the proposed dynamic training strategy.

**Evaluation Protocols.** We evaluate the learned representations with three protocols: standard linear finetuning (SLF), adversarial linear finetuning (ALF), and adversarial full finetuning (AFF). The former two protocols freeze the learned encoder and tune the linear classifier using cross-entropy loss with natural (SLF) or adversarial (ALF) samples, respectively. As for AFF, we adopt the pretrained encoder as weight initialization and train the whole classifier following ACL (Jiang et al., 2020).

### 5.1 BENCHMARKING THE PERFORMANCE OF DYNACL

**Robustness on Various Datasets.** In Table 1, we evaluate the robustness of sup-AT and self-AT methods on CIFAR-10, CIFAR-100, and STL-10. As we can see, DYNACL outperforms all previous self-AT methods in terms of robustness without using additional data[2]. Specifically, under the auto-attack benchmark, DYNACL brings a big leap of robustness by improving previous state-of-the-art self-AT methods by 8.84% (from 37.62% to 46.46%) on CIFAR-10, 4.37% (from 15.68% to 20.05%) on CIFAR-100, and 1.95% on STL-10 (from 45.26% to 47.21%). Furthermore, on CIFAR-10, DYNACL++ achieves even higher AA accuracy than the supervised vanilla AT (with heavily tuned default hyperparameters (Pang et al., 2020)). It is the first time that a self-supervised AT method outperforms its supervised counterpart.

**Robustness under Different Evaluation Protocols.** Table 2 shows that DYNACL and DYNACL++ obtain state-of-the-art robustness among self-AT methods across different evaluation protocols. In particular, under ALF settings, DYNACL++ outperforms all existing self-AT methods and even the sup-AT method. Furthermore, under AFF settings, DYNACL could improve the heavily tuned sup-AT baseline (Pang et al., 2020) by 1.58% AA accuracy (48.96% → 50.54%), and is superior to other pretraining methods. In conclusion, DYNACL and DYNACL++ have achieved state-of-the-art performance across various evaluation protocols.

**Performance under Semi-supervised Settings.** Following Jiang et al. (2020), we evaluate our proposed DYNACL++ under semi-supervised settings. Baseline methods are state-of-the-art semi-supervised AT method UAT++ [3] (Uesato et al., 2019), and self-supervised method ACL Jiang et al.

---

[2]Note that AdvCL incorporates extra data as it utilizes an ImageNet-pretrained model to generate pseudo labels. For a fair comparison, we replace AdvCL's ImageNet model with a model pretrained on the training dataset itself. We use AdvCL to denote this ablated version and refer to the original version as "AdvCL (+ImageNet)".

[3]UAT++ (Uesato et al., 2019) did not release the training code or pretrained ResNet-18 model. We copied the RA and SA results from ACL (Jiang et al., 2020).

Table 1: Comparison of supervised and self-supervised adversarial training methods on CIFAR-10, CIFAR-100, and STL-10. SA and AA stand for standard accuracy and robust accuracy under Auto-Attack (Croce & Hein, 2020). AdvCL (+ImageNet) uses additional ImageNet data. N/A: AdvCL does not provide ImageNet-based labels for STL-10.

| Pretraining Method | CIFAR-10 | | CIFAR-100 | | STL-10 | |
|---|---|---|---|---|---|---|
| | AA(%) | SA(%) | AA(%) | SA(%) | AA(%) | SA(%) |
| Sup-AT | 46.23 | 84.35 | 23.27 | 58.98 | 29.21 | 49.38 |
| RoCL | 26.12 | 77.90 | 8.72 | 42.93 | 26.51 | **78.19** |
| ACL | 37.62 | 79.32 | 15.68 | 45.34 | 33.24 | 71.21 |
| AdvCL | 37.46 | 73.23 | 15.45 | 37.58 | 45.26 | 72.11 |
| **DYNACL (ours)** | 45.04 | 77.41 | 19.25 | 45.73 | 46.59 | 69.67 |
| **DYNACL++ (ours)** | **46.46** | **79.81** | **20.05** | **52.26** | **47.21** | 70.93 |
| AdvCL (+ImageNet) | 42.57 | 80.85 | 19.78 | 48.34 | N/A | N/A |

Table 2: Performance comparison on CIFAR-10 with three evaluation protocols: SLF (standard linear finetuning), ALF (adversarial linear finetuning), and AFF (adversarial full finetuning).

| Pretraining Method | SLF | | ALF | | AFF | |
|---|---|---|---|---|---|---|
| | AA(%) | SA(%) | AA(%) | SA(%) | AA(%) | SA(%) |
| Sup-AT | 46.23 | 84.35 | 47.00 | 83.22 | 48.96 | 80.23 |
| RoCL | 26.12 | 77.90 | 29.69 | 75.62 | 45.02 | 78.51 |
| ACL | 37.62 | 79.32 | 40.91 | 76.57 | 49.46 | 82.11 |
| AdvCL | 37.46 | 73.23 | 37.28 | 73.15 | 48.58 | **82.31** |
| **DYNACL (ours)** | 45.04 | 77.41 | 45.72 | 72.87 | **50.54** | 81.84 |
| **DYNACL++ (ours)** | **46.46** | **79.81** | **47.95** | **78.84** | 50.31 | 81.94 |
| AdvCL (+ImageNet) | 42.57 | 80.85 | 42.54 | 79.41 | 49.97 | 83.27 |

(2020). The results are shown in Table 3. We observe that our DYNACL++ can demonstrate fairly good performance even when only 1% of labels are available.

## 5.2 EMPIRICAL UNDERSTANDINGS

In this part, we conduct comprehensive experiments on CIFAR-10 to have a deep understanding of the proposed DYNACL and DYNACL++. More analysis results are deferred to Appendix D.

**Ablation Studies.** We conduct an ablation study of our two major contributions (augmentation annealing in Section 4.1 and post-processing in Section 4.2) in Table 4. We can see that both annealing and post-processing can significantly boost the model's robustness, which demonstrates the superiority of our method.

**Robust Overfitting.** A well-known pitfall of sup-AT is *Robust Overfitting* (Rice et al., 2020): when training for longer epochs, the test robustness will dramatically degrade. Surprisingly, we find that this phenomenon does not exist in DYNACL. In Figure 9(a), we compare DYNACL and sup-AT for very long (pre)training epochs. We can see that sup-AT overfits quickly after the 100th epoch and suffers a 12% test robustness decrease eventually. Nevertheless, the test robustness of DYNACL keeps increasing *along the training process* even under 2000 epochs.

**Loss Landscape.** Previous studies (Prabhu et al., 2019; Yu et al., 2018; Wu et al., 2020) have shown that a flatter weight landscape often leads to better robust generalization. Motivated by that discovery, we visualize the loss landscape of ACL and DYNACL++ following (Li et al., 2018). Figures 9(b) and 9(c) show that DYNACL++ has a much flatter landscape than ACL, explaining the superiority of DYNACL++ in robust generalization.

**Training Speed.** We evaluate the total training time of self-AT methods with a single RTX 3090 GPU. Specifically, the total training time is 32.7, 105.0, 29.4, and 30.3 hours for ACL, AdvCL, DYNACL,

Table 3: Performance under semi-supervised settings on CIFAR-10. RA stands for robust accuracy under the PGD-20 attack.

| Label Ratio | UAT++ | | | ACL | | | DYNACL++ (ours) | | |
|---|---|---|---|---|---|---|---|---|---|
| | AA(%) | RA(%) | SA(%) | AA(%) | RA(%) | SA(%) | AA(%) | RA(%) | SA(%) |
| 1% labels | N/A | 30.46 | 41.88 | 45.65 | 50.46 | 74.76 | **46.95** | **51.30** | **76.77** |
| 10% labels | N/A | 50.43 | 70.79 | 45.47 | 50.01 | 75.14 | **48.56** | **53.00** | **78.34** |

Table 4: Ablation study two key designs of DYNACL++ on CIFAR-10. Baseline method is ACL (Jiang et al., 2020), and last two lines represent DYNACL and DYNACL++, respectively.

| Annealing | Post-processing | AA(%) | RA(%) | SA(%) |
|---|---|---|---|---|
| ✗ | ✗ | 37.62 | 40.44 | 79.32 |
| ✗ | ✓ | 43.24 (+5.62) | 45.48 (+5.04) | 77.40 (-1.92) |
| ✓ | ✗ | 45.04 (+7.42) | 48.40 (+7.96) | 77.41 (-1.91) |
| ✓ | ✓ | 46.46 (+8.84) | 49.21 (+8.77) | 79.81 (+0.49) |

and DYNACL++, respectively. Built upon ACL, DYNACL's dynamic strategy (augmentation and loss) introduces little computation overhead, and it spends even less time due to the stop gradient on the target branch. Moreover, since the post-processing only lasts a few epochs, DYNACL++ is also faster than ACL. In comparison, AdvCL is much slower and even triples the training time of ACL. These statistics show that our DYNACL and DYNACL++ are both efficient and effective.

**Effect of Annealing Schedule.** We evaluate DYNACL++ with different decay period $K$ in Table 5(a). We observe that our dynamic schedule significantly outperforms the constant one. Also, we notice that the step-wise annealing strategy performs slightly better. This may be because over-frequent data reloading (*i.e.,* distribution shift) may make the model fail to generalize well under each distribution.

**Effect of Reweighting Rate.** We also study the effect of annealing loss (Eq. 7) with different reweighting rates $\lambda$. Table 5(b) shows that compared to no annealing ($\lambda = 0$), the annealing loss ($\lambda > 0$) consistently improves both accuracy and robustness for around $1\%$, and the best robustness is obtained with a moderate reweighting rate $\lambda = 2/3$.

Table 5: Performance of DYNACL++ on CIFAR-10 with alternative annealing schedules and decay periods $K$ (in Eq. 5) (a), and with different reweighting rate $\lambda$ (in Eq. 7) (b).

| (a) decay schedule and decay period $K$ | | | | (b) reweighting rate $\lambda$ | | |
|---|---|---|---|---|---|---|
| schedule | $K$ | AA(%) | SA(%) | $\lambda$ | AA(%) | SA(%) |
| constant | - | 40.76 | 70.83 | 0 | 44.81 | 77.83 |
| linear | 1 | 45.77 | **80.26** | 1/3 | 45.63 | 79.55 |
| | 25 | 45.99 | 79.62 | 1/2 | 46.23 | 79.85 |
| | 50 | **46.46** | 79.81 | 2/3 | **46.46** | 79.81 |
| | 100 | 45.79 | 79.79 | 1 | 46.01 | **80.46** |

## 6 CONCLUSIONS

In this paper, we observe a dilemma about the augmentation strength in self-supervised adversarial training (self-AT) that either weak or strong augmentations degrade the model robustness through in-depth quantitative investigations. Going beyond the simple trade-off by selecting a medium constant augmentation strength, we propose to adopt a dynamic augmentation schedule that gradually anneals the strength from strong to weak, named DYNACL. Afterward, we devise a fast clustering-based post-processing technique to adapt the model for downstream finetuning, named DYNACL++. Experiments show that the proposed DYNACL and DYNACL++ successfully boost self-AT's robustness and even outperform its vanilla supervised counterpart on CIFAR-10 under Auto-Attack.

## ACKNOWLEDGEMENT

Yisen Wang is partially supported by the National Key R&D Program of China (2022ZD0160304), the National Natural Science Foundation of China (62006153), Open Research Projects of Zhejiang Lab (No. 2022RC0AB05), and Huawei Technologies Inc.

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

# A ALGORITHM DETAILS

## A.1 DETAILED CONFIGURATION OF DATA AUGMENTATIONS

We formulate the augmentation configuration $\mathcal{T}(s)$ with respect to augmentation strength $s$ in the following PyTorch-style code:

```python
from torchvision import transforms

def get_transforms(strength):
    rnd_color_jitter = transforms.RandomApply([transforms.ColorJitter(
            0.4 * strength, 0.4 * strength, 0.4 * strength, 0.1 *
    strength)], p=0.8 * strength)
    rnd_gray = transforms.RandomGrayscale(p=0.2 * strength)
    transform = transforms.Compose([
        transforms.RandomResizedCrop(
            32, scale=(1.0 - 0.9 * strength, 1.0)),
        # No need to decay horizontal flip
        transforms.RandomHorizontalFlip(p=0.5),
        rnd_color_jitter,
        rnd_gray,
        transforms.ToTensor(),
    ])
    return transform
```

## A.2 PSEUDO CODE OF DYNACL AND DYNACL++ ALGORITHM

---

**Algorithm 1:** DYNACL algorithm

---

**input** : Encoder $g$ and linear classification head $h$. Function of the augmentation set $\mathcal{T}(\cdot)$, number of pretraining epoch $T$, and post-processing epoch $T', T''$.

**output**: Pretrained and post-processed encoder $g$.

```
/* Phase 1:  Momentum contrastive pretraining with
   augmentation annealing (DYNACL)                         */
```
**1 forall** $t \in \{1, 2, \cdots, T\}$ **do**
**2**     **for** *sampled mini-batch* $\mathcal{X}$ **do**
**3**          augment samples from $\mathcal{X}$ by augmentation set $\mathcal{T}(s(t))$
**4**          $\mathcal{L} \leftarrow \mathcal{L}_{DYNACL}(g; t)$
**5**          update parameters in $g$ to minimize $\mathcal{L}$

```
/* Phase 2:  Pseudo label based post-processing (DYNACL++)  */
```
**6** Extract the normal route of the momentum encoder, and denote it by $g_c$
**7** $\{\hat{y}_i\} = \text{k\_means}(\{x_i\}; g_c)$
```
/* Post-processing phase 1, tune the head               */
```
**8** Extract and freeze the adversarial route of the momentum encoder, denote it by $g_a$
**9 forall** $epoch \in \{1, 2, \cdots, T'\}$ **do**
**10**      Standard-Training $(h \circ g_a, \{x_i\}, \{\hat{y}_i\})$
```
/* Post-processing phase 2, tune the whole model        */
```
**11** Unlock the parameters in $g_a$
**12 forall** $epoch \in \{1, 2, \cdots, T''\}$ **do**
**13**      TRADES $(h \circ g_a, \{x_i\}, \{\hat{y}_i\})$

---

Algorithm 1 demonstrates the pseudo-code of our proposed DYNACL and DYNACL++. Note that DYNACL++ adds a fast and simple post-processing phase to DYNACL and enjoys higher robustness.

## A.3 PIPELINE OF DYNACL AND DYNACL++

Figure 5 demonstrates the pipeline of the proposed DYNACL and DYNACL++. For DYNACL (upper part of the figure), we adopt our proposed augmentation annealing strategy and the momentum

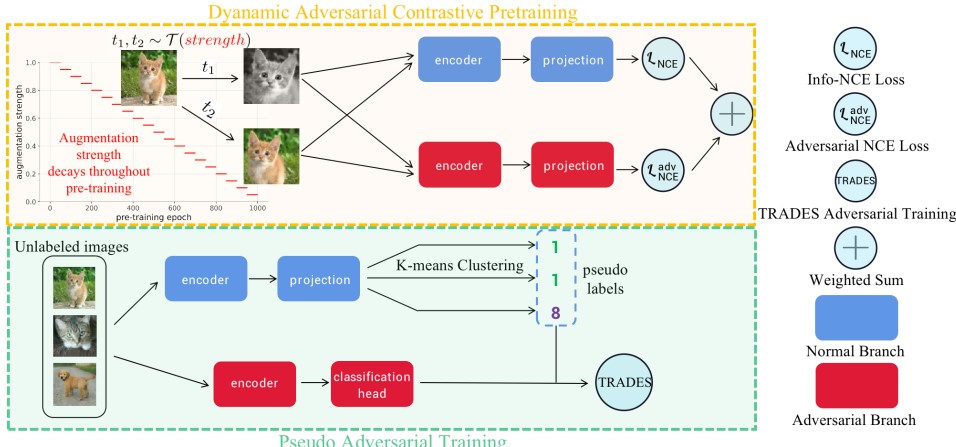

Figure 5: Pipeline of DYNACL.

technique (He et al., 2020) in the pretraining phase. Projection heads are added on the top of both branches of the ResNet backbone. For simplicity, this architecture is not explicitly shown in the figure. For DYNACL++, we add a simple and fast post-processing phase to DYNACL (bottom part of the figure). Specifically, we replace the projection head with a linear classification head and adopt our PAT algorithm. Note that both two phases require **NO** true labels.

## B DETAILS OF THE MEASUREMENT OF AUGMENTATION EFFECTS

### B.1 CALCULATION PROTOCOLS

**Calculation of minimal classwise distance.** We randomly augment each sample from the training set 50 times and calculate the distance in terms of $\ell_\infty$ norm in the input space. We believe that the input space class separability is a vital indicator of adversarial robustness. Typical adversarial perturbation budget $\varepsilon$ is defined in the input space, e.g., 8/255 under $\ell_\infty$-norm. Thus, given an augmented dataset $D$, whether the minimal input space classwise distance is smaller than $2 \cdot 8/255 = 16/255$ decides whether this dataset is adversarially separable, i.e., whether there exists a classifier that achieves 100% training robust accuracy.

**Calculation of Maximum Mean Discrepancy (MMD).** Denote the whole training set by $\mathcal{X}_{tr}$ and the whole test set by $\mathcal{X}_{ts}$. We first transform each set into a batch of tensors and then flatten them into two dimensions. Finally, we adopt Radical Basis Function as kernel function with bandwidth $10, 15, 20, 50$ to calculate the MMD between $\mathcal{X}_{tr}$ and $\mathcal{X}_{ts}$.

### B.2 LATENT SPACE DISTRIBUTION GAP AND MINIMAL CLASSWISE DISTANCE

In Section 3.1, we have demonstrated the input space distribution gap between the training set and test set and found that stronger data augmentation will result in a wider distribution gap. To further justify our findings, we calculate the statistics using the distance of sample features in the latent space. Specifically, we utilize the Perceptual Similarity proposed in (Zhang et al., 2018) to calculate latent space distance.

**Latent Space Distribution Gap.** Specifically, for MMD, given perceptual similarity function $f(x, y)$ which takes two images $x, y$ as inputs, the kernel function in MMD is defined as $1 - f(x, y)$. Note that the perceptual similarity is lower for similar images and 0 for identical images. Figure 6(a) demonstrates the latent space MMD between the augmented training set and clean test set. Similar to input space MMD, stronger data augmentation will bring a bigger distribution gap, thus harming robustness generalization.

**Latent Space Minimal Classwise Distance.** Similarly, we can calculate the classwise distance using latent space distance. Specifically, we augment each sample from the training set 20 times and use

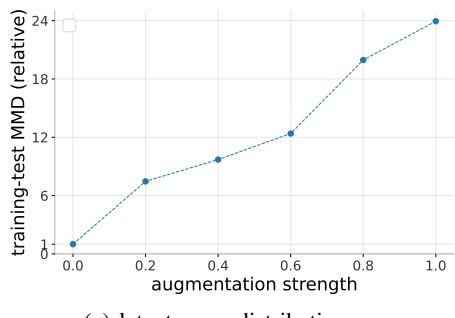
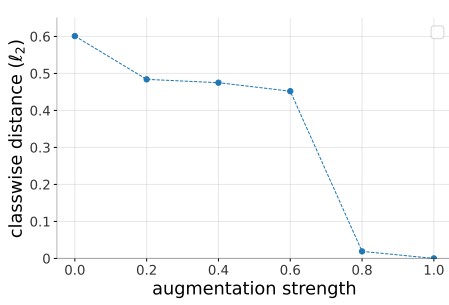

(a) latent space distribution gap

(b) latent space minimal classwise distance

Figure 6: (a) Latent space relative MMD between the augmented training set and clean test set. (b) Latent space minimal classwise distance on the training set.

a pretrained AlexNet to extract its latent space features. After normalization, We calculate the $\ell_2$ distance between augmented samples from different classes and record the minimum. The results are shown in Figure 6(b). We can see that the latent classwise distance will also become significantly smaller under stronger augmentations (larger $s$), which generally agrees with the trend in the input space.

## C  EXPERIMENTAL DETAILS

### C.1  PRETRAINING SETTINGS

Except that we adopt the momentum backbone and stop-gradient operation (Grill et al., 2020), we follow ACL on other pretraining configurations. Specifically, we train the backbone for 1000 epochs with LARS optimizer and cosine learning rate scheduler. The projection MLP has 3 layers. For DYNACL pretraining, we set the decay period $K = 50$, and reweighting rate $\lambda = 2/3$. For DYNACL++, we generate pseudo labels for training data by K-means clustering. First, we tune the classification head only by standard training for 10 epochs, then we tune both the encoder and the classification head by TRADES (Zhang et al., 2019) for 25 epochs.

### C.2  EVALUATION PROTOCOLS

As for SLF and ALF we train the linear classifier for 25 epochs with an initial learning rate of $0.01$ on CIFAR-10 and $0.1$ on CIFAR-100 and STL-10 (decays at the 10th and 20th epoch by 0.1) and batch size 512. For AFF, we employ the TRADES (Zhang et al., 2019) loss with default parameters, except that in DYNACL and DYNACL++, we feed the clean samples to the normal encoder when generating adversarial perturbations. Finetuning lasts 25 epochs with an initial learning rate of $0.1$ (decays by 0.1 at the 15th and 20th epoch) and batch size 128. For all baselines and DYNACL, we report the last result in the evaluation phase.

In the original implementation of AdvCL (Fan et al., 2021), they incorporate extra data from ImageNet to train a feature extractor based on SimCLR. For a fair comparison, we replace AdvCL's ImageNet-pretrained model with a CIFAR-pretrained model. This model is trained following the default hyperparameters and official implementations and reaches a clean accuracy of $88.15\%$ on CIFAR-10 and $65.78\%$ on CIFAR-100.

## D  ADDITIONAL EXPERIMENTS

### D.1  TRAINING DYNAMICS

Figure 7 demonstrates the training dynamics of ACL, DYNACL, and supervised-AT (all with cosine learning rate schedule). We observe that the robust accuracy (PGD-20) of both DYNACL and ACL increases continuously throughout the training process, while that of sup-AT suffers a significant decrease in the later period of training.

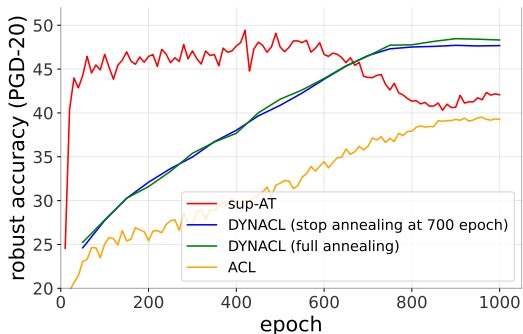

Figure 7: Training dynamics of ACL, DYNACL, and supervised AT.

Figure 7 also depicts the effectiveness of annealing the augmentation strength $s$ to 0. To further examine this point, we consider an alternative schedule, where we threshold the augmentation strength when it reaches $s = 0.3$ at the 700-th epoch, and adopt a constant augmentation strength $s = 0.3$ afterwards. Table 6 shows that the thresholding strategy performs a little bit worse than the original strategy.

Table 6: Comparison between the threshold strategy and the original strategy.

|  | AA(%) | SA(%) |
| --- | --- | --- |
| original $(1 \rightarrow 0)$ | **46.46** | 79.81 |
| threshold at 0.3 | 45.89 | **80.14** |

## D.2 DATA AUGMENTATION STRENGTH IN SUP-AT

Popular sup-AT methods (Zhang et al., 2019; Pang et al., 2020) typically adopt very weak data augmentations, including only horizontal flip and padding-crop. As shown in Figure 8, aggressive data augmentations are detrimental to the model's performance. For self-AT, this phenomenon also holds, and the corresponding results are shown in 4(c).

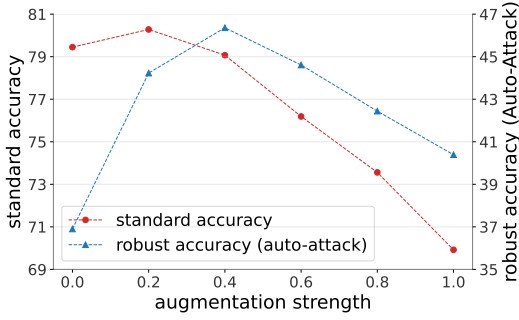

Figure 8: Performance of sup-AT with respect to augmentation strength $s$

## D.3 ABLATION OF MOMENTUM ENCODER AND STOP GRADIENT OPERATION

Here we conduct an ablation study on the momentum encoder and stop-gradient operation that is additional adopted by DYNACL. We can see without both techniques, DYNACL still achieves very high AA robustness (44.9%) that is significantly better than ACL (Jiang et al., 2020). As for the two techniques, we notice that the stop gradient alone has limited improvement, also it helps improve training efficiency. The momentum encoder, instead, help improves the standard accuracy by nearly 2%, while attains similar AA robustness to the vanilla DYNACL (ACL + dynamic strategy).

Table 7: Ablation study of momentum backbone and stop gradient operation on the CIFAR-10 dataset. Our DYNACL (last line) uses both momentum encoder and stop gradient operation following (Grill et al., 2020). Baseline method (first line) is ACL (Jiang et al., 2020).

| Dynamic Strategy | Stop Gradient | Momentum | AA(%) | RA(%) | SA(%) |
|:---:|:---:|:---:|:---:|:---:|:---:|
| × | × | × | 37.62 | 40.44 | **79.32** |
| ✓ | × | × | 44.92 | **48.65** | 75.76 |
| ✓ | ✓ | × | 45.02 | 48.57 | 75.00 |
| ✓ | ✓ | ✓ | **45.04** | 48.40 | 77.41 |

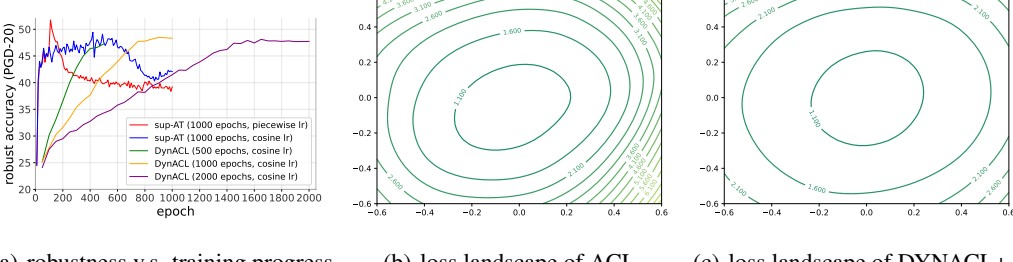

(a) robustness v.s. training progress      (b) loss landscape of ACL      (c) loss landscape of DYNACL++

Figure 9: (a) Comparison of Robust Overfitting between sup-AT and self-AT (DYNACL) with different pretraining schedules (without post-processing) on CIFAR-10. (b), (c): loss landscape visualization of ACL and DYNACL++. Note that DYNACL++ enjoys a flatter loss landscape compared with ACL.

## D.4 ROBUST OVERFITTING AND LOSS LANDSCAPE

We provide illustrations on robust overfitting and loss landscape in Figure 9. Please refer to Sec. 5.2 for detailed discussions.

## E THEORETICAL JUSTIFICATION

In this section, following the theoretical analysis of adversarial training (Sinha et al., 2017), we provide a solid theoretical justification on our analysis. Specifically, we show that a large training-test distribution gap caused by aggressive data augmentations will indeed induce larger test errors.

**Setup.** Consider an input data space $\mathcal{X}$ and a bounded loss function $l(x;\theta)$ satisfying $|l(x;\theta)| \leq M$. For the ease of theoretical exposure, we consider the Wasserstein distance for measuring the distribution gap. In particular, the Wasserstein distance between $P$ and $Q$ is

$$W_c(P,Q) := \inf_{M \in \Pi(P,Q)} \mathbb{E}_M \left[ c\left(X, X'\right) \right] \tag{8}$$

where $\Pi(P,Q)$ denotes the couplings of $P$ and $Q$, and $c : X \to X$ defines a non-negative, lower semi-continuous cost function satisfying $c(x,x) = 0, \forall\, x \in \mathcal{X}$. Accordingly, we adopt a robust surrogate objective $\phi_\gamma(\theta; x_0) := \sup_{x \in \mathcal{X}} \{\ell(\theta; x) - \gamma c(x, x_0)\}$, which allows adversarial perturbations of the data $x$ in the Wasserstein space, modulated by the penalty $\gamma$. Considering the equivalence of norms in the finite-dimensional space, our discussion here in the Wasserstein space also sheds light on $\ell_p$-norm bounded adversarial training.

Suppose the original data distribution (without augmentation) is $P_{\text{test}}$, and the augmented data distribution is $P_{\text{aug}}$. Denote the empirical augmented training data distribution as $\hat{P}_{\text{aug}}$ with $n$ samples. Below, we are ready to establish theoretical guarantees on the out-of-distribution (OOD) generalization of *robust training* on the augmented data, *i.e.,* $\hat{P}_{\text{aug}}$ to the original test data distribution $\hat{P}_{\text{test}}$.

**Theorem 1** *For any and a fixed $t > 0$, the following inequality holds with probability at least $1 - e^{-t}$, simultaneously for all $\theta \in \Theta, \rho \geq 0, \gamma \geq 0$:*

$$\sup_{P: W_c(P_{\text{test}}, P_{\text{aug}}) \leq \rho} \mathbb{E}_{P_{\text{test}}}[\ell(\theta; X)] \leq \gamma \rho + \mathbb{E}_{\widehat{P}_{\text{aug}}}[\phi_\gamma(\theta; X)] + \epsilon_n(t). \tag{9}$$

*Here, $\epsilon_n(t) := \gamma b_1 \sqrt{\frac{M}{n}} \int_0^1 \sqrt{\log N\left(\mathcal{F}, M\epsilon, \|\cdot\|_{L^\infty(\mathcal{X})}\right)} d\epsilon + b_2 M \sqrt{\frac{t}{n}}$, where $\mathcal{F}$ is the hypothesis class, $N(\mathcal{F}, \varepsilon, \|\cdot\|)$ is the corresponding cover number, and $b_1$ and $b_2$ are numerical constants.*

**Analysis.** From Theorem 1, we can see that $\rho$ controls the distribution gap between the original data distribution and the augmented data distribution, and a larger $\rho$ indeed introduces a larger error term in the upper bound of the test performance. Specifically, when $\rho$ is very large, *e.g.,* with aggressive augmentations, the test performance guarantees to be poor. This shows that theoretically, a too large training-test distribution gap introduced by aggressive augmentation will indeed contribute to worse generalization, which echoes with our empirical investigation in Section 3.

Consequently, if we gradually anneal the augmentation strength $s$ from 1 to 0, as we have done in the proposed DYNACL, the distribution discrepancy $\rho$ will gradually shrink (as it is nearly 0 when $s = 0$ at last), leading to a smaller upper bound. Thus, our augmentation annealing strategy will indeed help bridge the distribution gap and improve generalization to test data.

### E.1 PROOF OF THEOREM 1

We first state the following lemma from Sinha *et al.* (Sinha et al., 2017) that gives a duality result between the objectives on any two distributions $P$ and $Q$.

**Lemma 2** *Let $\ell : \Theta \times \mathcal{X} \rightarrow \mathbb{R}$ and $c : \mathcal{X} \times \mathcal{X} \rightarrow \mathbb{R}_+$ be continuous. Let $\phi_\gamma(\theta; z_0) = \sup_{z \in \mathcal{X}} \{\ell(\theta; z) - \gamma c(z, z_0)\}$ be the robust surrogate (2 b). For any distribution $Q$ and any $\rho > 0$,*

$$\sup_{P: W_c(P, Q) \leq \rho} \mathbb{E}_P[\ell(\theta; X)] = \inf_{\gamma \geq 0} \{\gamma \rho + \mathbb{E}_Q[\phi_\gamma(\theta; X)]\}$$

*and for any $\gamma \geq 0$, we have*

$$\sup_P \{\mathbb{E}_P[\ell(\theta; X)] - \gamma W_c(P, Q)\} = \mathbb{E}_Q[\phi_\gamma(\theta; X)].$$

Applying Lemma 2 to our problem, we have the following deterministic result

$$\sup_{P_{\text{test}}: W_c(P_{\text{test}}, P_{\text{aug}}) \leq \rho} \mathbb{E}_{P_{\text{test}}}[\ell(\theta; X)] \leq \gamma \rho + \mathbb{E}_{P_{\text{aug}}}[\phi_\gamma(\theta; X)]$$

for all $\rho > 0$, distributions $P_{\text{aug}}$, and $\gamma \geq 0$. Next, we show that $\mathbb{E}_{\widehat{P}_{\text{aug}}}[\phi_\gamma(\theta; X)]$ concentrates around its population counterpart at the usual rate. Also remind that we have that $\phi_\gamma(\theta; z) \in [-M, M]$, because $-M \leq \ell(\theta; x) \leq \phi_\gamma(\theta; x) \leq \sup_x \ell(\theta; x) \leq M$. Thus, the functional $\theta \mapsto F_n(\theta)$ satisfies bounded differences, and applying standard results on Rademacher complexity and entropy integrals gives the result.

