# OpenReview forum: "Rethinking the Effect of Data Augmentation in Adversarial Contrastive Learning"
_ICLR.cc/2023/Conference — ICLR 2023 poster_

### Official Review · Reviewer_Kni5 · 2022-10-22

**Confidence:** 4
**Correctness:** 3
**Technical Novelty And Significance:** 2
**Empirical Novelty And Significance:** 3
**Recommendation:** 6

**Clarity, Quality, Novelty And Reproducibility:**

The paper is well-organized and written. I can easily follow most of the parts. The novelty of the proposed method seems to be fair. This paper has good reproducibility since the authors have provided the algorithm in the appendix and illustrated the hyper-parameters in detail.

**Strength And Weaknesses:**

Strength
+ This paper provides an interesting and insightful investigation of the effects of data augmentation in ACL. The empirical investigation validates that stronger data augmentation can be harmful to ACL.
+ The proposed method is compatible with variants of ACL method and has no extra computational consumption. The experiments validate DynACL can significantly improve adversarial robustness compared to previous self-supervised adversarial training methods.

Weaknesses
- The investigation totally depends on the empirical phenomenon. Say the classwise distance. The paper only shows the phenomenon. But it seems that the paper does not provide more (theoretically) analyses of the reason for this phenomenon.
- The proposed dynamic scheduling is somewhat naïve. And, it is strange that the training procedure still uses the most aggressive data augmentation at the final stage, which has been pointed out as harmful to ACL.



**Summary Of The Paper:**

This paper investigates the effects of data augmentation in adversarial contrastive learning (ACL). The authors show that stronger data augmentation can result in a larger distribution gap between training data and test data, and a smaller classwise distance. Therefore, the authors propose a piecewise decay augmentation schedule for ACL, namely DynACL. The empirical results validate that DynACL can achieve comparable and even better adversarial robustness than supervised adversarial training (Madry et al., Towards deep learning models resistant to adversarial attacks, 2017).

**Summary Of The Review:**

Overall, the empirical results solidly support the effectiveness of the proposed method. However, the analysis of some investigation (e.g., why a more aggressive data augmentation can incur a smaller classwise distance?) is not very clear. Therefore, I tend to borderline accept.

---

> ### Author Response · Authors · 2022-11-13
> **Response to Reviewer Kni5**
>
> We express our sincere gratitude to Reviewer Kni5 for appreciating the soundness of the investigation and the empirical success of our work. We address your main concerns below.
>
> ---
> **Q1.** The investigation totally depends on the empirical phenomenon, e.g., lack of theoretical reason for the shrink of classwise distance.
>
> **A1.** In fact, we have included a **theoretical justification of the large distribution gap in Appendix E.1**. Leveraging the distributionally robust optimization techniques, we theoretically prove that a large distribution gap between training and test data will hurt generalization and induce large test errors, which is consistent with our empirical observation in Fig 4c.
>
> **Newly added theoretical justifications.** For completeness, we further add a theoretical explanation on **classwise inseparability (Appendix E.2)**, where we theoretically prove that strong data augmentations will shrink the classwise distance and even make data inseparable, which is consistent with our empirical observation in Fig 2.
>
> Our dynamic schedule gradually **decreases the augmentation strength from 1 to 0** and approaches the test distribution at last (Eq. 8). Therefore, according to the theoretical results above, **our method can theoretically maintain a small training-test distribution gap and a large classwise distance** at the final training stage.
>
> ---
> **Q2.** The proposed dynamic scheduling is somewhat naïve. And it is strange that the training procedure still uses the most aggressive data augmentation at the final stage, which has been pointed out as harmful to ACL.
>
> **A2.** We are afraid that you might misunderstand our method here. In our proposed dynamic schedule, the augmentation strength $s$ is gradually annealing from 1 to 0 along the training process (see Eq. 5 for a formal expression and Fig. 3 for an illustration). Therefore, at the final training stage, we arrive at $s=0$, the mildest augmentation that matches the test data distribution, which is the opposite of your description. We are not sure we have fully understood your point here, and please let us know if there is more to clarify.
>
> ---
> Thanks for your encouraging and constructive comments. Hope our explanations and additional theoretical discussions can address your concerns. We are happy to take your further questions during the rebuttal stage.

---

### Official Review · Reviewer_UQhW · 2022-10-24

**Confidence:** 4
**Correctness:** 3
**Technical Novelty And Significance:** 3
**Empirical Novelty And Significance:** 3
**Recommendation:** 6

**Clarity, Quality, Novelty And Reproducibility:**

The paper is well organized, but the presentation has minor details that could be improved.
The technicality of the paper appears to be sound, and the theoretical knowledge is explicit.
The idea to propose a dynamic augmentation strategy along the training to improve the robustness of self-AT is novel.
The code is not available, but the experimental details are sufficient. It doesn't seem difficult to reproduce.


**Strength And Weaknesses:**

Strength:

a)	The paper reveals the reason behind the robustness gap between self-AT and sup-AT and investigates the effect of data augmentation strategy on self-AT.

b)	The proposed DynACL shows a significant improvement in clean accuracy and robustness over existing self-AT methods.

c)	The DynACL is also more computationally efficient.

d)	This paper is well written.

Weaknesses:

a)	In Table 1, the standard accuracy (SA%) of DynACL on STL-10 is significantly lower than other self-AT methods, which is not explained in the paper.

b)	The metric RA(%) in Table 2 is not explained.


**Summary Of The Paper:**

This paper first reveals a dilemma about the augmentation strength that either strong or weak data augmentations are harmful to self-supervised adversarial training (self-AT). To resolve the dilemma,  the paper proposes a simple remedy named DynACL (Dynamic Adversarial Contrastive Learning). DynACL adopts a dynamic augmentation schedule along the training process that gradually anneals the strength from strong to weak, then adopts a fast post-processing stage for adapting it to downstream tasks.

**Summary Of The Review:**

The paper is well written, the contribution is clearly presented and the experimental results are thoroughly executed. I would tend to accept this paper.

---

> ### Author Response · Authors · 2022-11-13
> **Response to Reviewer UQhW**
>
> We express our sincere gratitude to Reviewer UQhW for appreciating our work's clarity, soundness, and effectiveness. We address your main concerns below.
>
> ---
>
> **Q1.** In Table 1, the standard accuracy (SA%) of DynACL on STL-10 is significantly lower than other self-AT methods, which is not explained in the paper.
>
> **A1.** Observing Table 1 (quoted below), we can see that DynACL's SA (69.67%) is only slightly lower than backbone ACL (71.21%) and AdvCL (72.11%), while it also has higher AA robustness than the two (**+13.3% AA vs. ACL, +1.3% AA vs. AdvCL**). Here, RoCL has much higher SA (78.19%) than all the rest, mainly because **its AA robustness (26.51%) is much lower than others**. In AT, there is generally a trade-off between accuracy and robustness [1,2], showing that better robustness often comes with lower standard accuracy. This explains the higher SA of RoCL and the lower SA of DynACL.
>
> | Method | AA% | SA% |
> | - | - | - |
> | RoCL | 26.51 | 78.19|
> | ACL | 33.24 | 71.21|
> | AdvCL | 45.26 | 72.11|
> | DynACL (ours) | 46.59 | 69.67|
> | DynACL++ (ours) | **47.51** | 70.93|
>
> Further reflecting on the slight drop of SA in DynACL could be due to the weaker augmentations adopted by DynACL at later training stages, indicating that STL-10 requires stronger augmentations to learn good representations. Nevertheless, the post-processing step in DynACL++ largely alleviates the SA degradation of DynACL. As a result, DynACL++ has **2\% higher AA** and only 1.2% lower SA than AdvCL, showing the superiority of our proposed method on STL-10. On other datasets like CIFAR-10 and CIFAR-100, the advantages of DynACL are more notable as it improves both SA and AA significantly over AdvCL.
>
> [1] Robustness may be at odds with accuracy. In ICLR, 2019. https://openreview.net/pdf?id=SyxAb30cY7
>
> [2] Theoretically principled trade-off between robustness and accuracy. In ICML, 2019. http://proceedings.mlr.press/v97/zhang19p/zhang19p.pdf
>
> ---
> **Q2.** The metric RA(%) in Table 2 is not explained.
>
> **A2.** Are you referring to Table 3 here? Since there is no RA in Table 2, but in Table 3. Here, RA stands for Robust Accuracy obtained under the PGD-20 attack. This protocol is a common metric for robustness evaluation, faster to compute, but is also known to be less reliable than AutoAttack (AA) robustness. We include RA here mainly for comparison since the baseline UAT++ only reported results under PGD-20 instead of AA. We have explained RA in the caption in the revised version.
>
> ---
> Thanks again for your careful reading and constructive review. We have revised our paper according to your suggestions. We are happy to take your further questions during the rebuttal stage.

---

> > ### Comment · Reviewer_UQhW · 2022-11-25
> > **Reply**
> >
> > Many thanks for the response.
> > My concerns have been solved.

---

### Official Review · Reviewer_umnn · 2022-10-25

**Confidence:** 5
**Correctness:** 4
**Technical Novelty And Significance:** 2
**Empirical Novelty And Significance:** Not applicable
**Recommendation:** 8

**Clarity, Quality, Novelty And Reproducibility:**

**Clarity:** The paper is easy to understand and well-written.

**Quality:** The presentation of the paper is good and well-organized.

**Novelty:** The dynamic gradual design of the data augmentation seems not very novel to me but it is simple and effective.

**Reproducibility:** The paper has well reproducibility which elaborates well on the details.

**Strength And Weaknesses:**

**Strength**

- This paper proposes a simple and effective scheduling method for selfsup-AT which could surprisingly improve the robustness.
- The paper is well-written and shows convincing motivations.
- This paper suggests novel aspects of the relationship between augmentation and adversarial training in selfsup-AT.
- This paper demonstrates extensive experimental results including diverse ablation experiments and semi-supervised settings.

**Concerns and Questions**
- I hope the authors describe the difference between the DynACL++ and AdvCL post-processing which pseudo labels the unlabeled examples with k mean clustering. I think it is overemphasized claims that fast post-processing is a novel idea.
- Does proposed technique also can be adapted to RoCL? And also boost the performance of this method too?
- Does dynamic scheduling also benefit the performance of vanilla simCLR?
- The proposed dynamic scheduling which gradually decreases the strength of the augmentation seems a somewhat empirical approach seems bit simple approach without any convincing motivation or evidence. It is understandable that we should control the strength of augmentation but I am not sure proposed method is the most effective design of data augmentation. Because if there is some standard to find the adequate strength of the augmentation at a certain training stage of self-AT (i.e., classwise distance, or MMD), an adaptive strength controller could show better robustness than the gradual scheduling.
- I am not quite sure this approach could be generalized to different architectures, or different frameworks. Since the approach is a naive approach, hyperparameter search may be always needed for optimal performance. And the current gradual scheduling could not be the best option for some circumstances (where may need a longer step size). I think adaptive data augmentation scheduling [1,2] could also be applied to self-AT which could improve the current approach more intuitively and make the approach to be more universal to diverse circumstances.

[1] ADAAUG: LEARNING CLASS- AND INSTANCE- ADAPTIVE DATA AUGMENTATION POLICIES, ICLR 2022

[2] MetaAugment: Sample-Aware Data Augmentation Policy Learning, AAAI 2021

**Summary Of The Paper:**

This paper suggests dynamic adversarial contrastive learning which gradually anneals from a strong augmentation to a weak augmentation. Further, the authors propose fast post-processing stage for adapting it to classification tasks which boost the robustness. From this simple and effective strategy, DynACL reduces the gap between supervised AT and selfsup AT.

**Summary Of The Review:**

Overall, I recommend acceptance to this paper. This is well written and has intuitive motivation with empirical results.

---

> ### Author Response · Authors · 2022-11-13
> **Response to Reviewer umnn (3/3)**
>
> **Q6.** Since the approach is naive, a hyperparameter search may always be needed for optimal performance. And the current gradual scheduling could not be the best option for some circumstances (where it may need a longer step size).
>
>
> **A6.** As shown in Table 5a (quoted below), even **a vanilla choice of the step size K** (e.g., K=1) **can bring a 5% improvement in AA robustness** over ACL (constant schedule). Meanwhile, our method is relatively robust to the choice of K, as ranging K from 1 to 100 only causes around a 1% change in AA. These results suggest that **the proposed dynamic schedule is the key to improvement, and the specific type of schedules does not affect much**.
>
>
> #### *Table 5(a). AA Robustness on CIFAR-10 using dynamic schedules with different K*
> |constant | K=1 | K=25 | K=50 | K=100 |
> |- | - | -| -| -|
> |40.76 | 45.77 | 45.99 | 46.46 | 45.79|
>
> **Robustness to Hyperparameters.** In our experiments, we adopt **the same K=50 for all datasets, backbones, and SSL frameworks**, including CIFAR-10, CIFAR-100, and STL-10, and **this default setting yields state-of-the-art robustness on all datasets** (Table 1 and results above). Therefore, K=50 is a stable and effective default parameter choice, and **we do not *have to* tune K to work on a new dataset, architecture, or framework**.
>
> ---
> **Q7.** I think adaptive data augmentation scheduling (AdaAug) [1] could also be applied to self-AT, which could improve the current approach more intuitively and make the approach more universal to diverse circumstances.
>
> **A7.** Thanks for pointing out these related works. As discussed above, our method applies to various scenarios with the same augmentation schedule. Following your suggestions, we also compare it with the adaptive augmentation method (AdaAug) you mentioned. The results are shown below.
>
> #### *Comparison between AdaAug [1] and our methods on CIFAR-10*
> | | AA(%) | SA(%) |
> | - | - | - |
> | ACL | 37.62 | 79.32 |
> | AdaAug [1] | 37.91 | 80.57 |
> | DynACL (ours) | **45.04** | 77.41 |
> | DynACL++ (ours) | **46.46** | 79.81 |
>
> **Performance.** We find that AdaAug can bring slight improvement over ACL, but its robustness is still much inferior to our DynACL and DynACL++ (8.5\% lower AA). Therefore, existing adaptive augmentations might not be as effective as our proposed dynamic annealing schedule. We believe the reason could be that they do not explicitly consider the dynamic trade-off between accuracy and robustness.
>
> **Efficiency.** Besides, DynACL enjoys being **simple and easy to implement**: it only requires modifying a few lines of code and brings **no additional parameter and training cost**. However, learning adaptive augmentations often introduce **extra modules and training expenses** to meta-learn a suitable strategy.
>
> Therefore, DynACL is still advantageous due to its simplicity and efficiency. Meanwhile, devising a more effective adaptive strategy is a promising direction, and we will explore it in future work. We have added this discussion in the revision (**Appendix D**).
>
> [1] AdaAug: Learning Class- and Instance-adaptive Data Augmentation Policies. ICLR 2022.
>
> ---
> Thanks for your insightful and constructive comments. Hope our explanations and additional experiments can address your concerns. Please let us know if there is more to clarify. We are happy to take your further questions during the rebuttal stage.

---

> ### Author Response · Authors · 2022-11-13
> **Response to Reviewer umnn (2/3)**
>
> (continued)
>
> We can see that **smaller augmentation strengths ($s<1$) always hurt model accuracy in self-ST** (self-supervised standard training). As a result, **a dynamic schedule that decreases augmentation strength will naturally hurt self-ST**. This is consistent with our analysis in **Introduction** and **Sec 3**, that strong augmentations are helpful for accuracy while being harmful to robustness. In this regard, the dynamic strategy proposed in this work is mainly designed to **improve robustness** instead of accuracy. Other dynamic schedules might also help self-ST, but since it is not the focus of this paper, we leave it for future work.
>
> ---
> **Q4.** Questions on the design of dynamic schedules.
>
> **A4.** We address each of your concerns point by point.
> > The proposed dynamic scheduling seems a somewhat empirical approach seems bit simple approach without any convincing motivation or evidence.
>
> **Motivation from empirical observation.** The motivation of our dynamic schedule is explained in **Sec 3.2 & 4.1**. On the one hand, **Fig 4a & 4b** show that a strong augmentation strength causes **large distribution gap and class inseparability** that hurt adversarial robustness, and a simple fix to this problem is to **decrease the augmentation strength** to a small value (Fig 4a, 4b). On the other hand, Fig1b shows that **strong augmentations** are also necessary for obtaining good representations with contrastive learning. To **meet the two requirements at the same time,** we propose a dynamic schedule that applies strong augmentations at the beginning and gradually anneals it to a small value to mitigate the distribution gap at last.
>
> **Existing theoretical justification on large distribution gap.** In **Appendix E.1**, leveraging the distributionally robust optimization techniques, we theoretically prove that a large distribution gap between training and test data will hurt generalization and induce large test errors, which is consistent with our empirical observation in Fig 4c.
>
> **Newly added theoretical justifications.** For completeness, we further add a theoretical explanation on **classwise inseparability (Appendix E.2)**, where we theoretically prove that strong data augmentations will shrink the classwise distance and even make data inseparable, which is consistent with our empirical observation in Fig 2.
>
> Further, as our proposed dynamic schedule gradually **decreases the augmentation strength from 1 to 0** and approaches the test distribution at last (Eq. 8). Therefore, according to the theoretical results above, **our method can theoretically maintain a small training-test distribution gap and a large classwise distance** at the final training stage.
>
> > Is there some standard to find the adequate strength of the augmentation at a certain training stage of self-AT (i.e., classwise distance or MMD)? Will an adaptive strength controller show better robustness than gradual scheduling?
>
> Indeed, we agree that a more adaptive strategy could yield better performance. Nevertheless, finding such a metric is challenging. The tricky thing here is that **we should dynamically balance the two requirements (representative power (need strong aug) and adv robustness (need weak aug))**. The metrics you mention, classwise distance and MMD, only consider the effect on robustness alone, and pursuing robustness alone may degrade representation power. As for the former, standard evaluation metrics (like linear probing accuracy) require labels to compute, which are unavailable during training. In comparison, our proposed annealing strategy is simple, easy to use, and robust to hyperparameter choices (Table 5a). As it achieves significant improvements on benchmark datasets, it could serve as a simple and strong baseline for the future development of more adaptive strategies.
>
>
> ---
> **Q5.** Generalization to different architectures and different frameworks.
>
> **A5.** Following your suggestions, we also evaluate ACL and our DynACL++ on a) a **different architecture, AlexNet**, and b) a **different SSL framework, SimSiam**, which is a non-contrastive method based on latent bootstrapping. We adopt the **default hyperparameters** without further tuning. The tables below show that DynACL++ **improves ACL by 12.14\% AA robustness on AlexNet**and **4.04\% AA robustness on SimSiam**, indicating that our method is effective and generalizable across different architectures and frameworks.
>
> #### *a) Generazalition to a new architecture AlexNet  on CIFAR-10*
> | | AA(%) | SA(%) |
> | - | - | - |
> | ACL | 11.79 | 54.87 |
> | DynACL++ (ours) | **23.93** | 56.61 |
>
> #### *b) Generalization to a new SSL framework SimSiam (bootstrapping methods) on CIFAR-10*
> | | AA(%) | SA(%) |
> | - | - | - |
> | Adversarial SimSiam | 32.02 | 67.05 |
> |**Adversarial SimSiam + DynACL++ (ours)** | **36.09** | 66.61 |

---

> > ### Comment · Reviewer_umnn · 2022-11-15
> > **Thank you for your detailed response.**
> >
> > Thank you for your detailed response.
> >
> > My initial concerns are almost resolved.
> > **I agree that your work shows extensive experiments with surprisingly good performance.**
> > Therefore, I raise my score to 8.
> >
> > I now only have a few questions about experimental details which is just a simple question not very related to my concerns.
> > In your response, what does RoCL backbone means?
> > Did you use the same attack loss as RoCL and training loss with RoCL and apply your dynamic augmentation scheduling or just use the backbone architecture of RoCL and use ACL attack loss and ACL training with Dynamic augmentation scheduling?
> >
> > Your motivation and the theoretical justification show that less distribution difference will lead to better generalization. If so, why strong augmentation is needed? I feel this result may contradict to the base intuition of self-supervised learning where learning agreement between strong augmentation leads to better generalization. Why not we just use vanilla augmentation from the first time to train the adversarial self-supervised learning model where the model will always have a small training-test gap? I am just curious about the authors’ thoughts.
> >
> > As I know, SimSiam does not use contrastive learning loss to train the model. How did you apply DynACL on top of SimSiam? And is the ACL in the A5. table b typo of SimSiam?
> >
> > Thanks again for the detailed response and I also look forward to discussing this paper with other reviewers and ACs.

---

> > > ### Author Response · Authors · 2022-11-17
> > > **Thanks and Further Response**
> > >
> > > Thanks for your quick reply and for appreciating our response! We further address your questions as follows:
> > >
> > > ---
> > > **Q1.** What does RoCL backbone mean? Did you use the same attack loss as RoCL and training loss with RoCL and apply your dynamic augmentation scheduling or just use the backbone architecture of RoCL and use ACL attack loss and ACL training with Dynamic augmentation scheduling?
> > >
> > > **A1.** Here, we only modify the augmentation schedule from their constant schedule to our dynamic annealing schedule. For fair comparison, we adopt all default training configurations of RoCL in their official code, including their attack and training objectives.
> > >
> > > ---
> > > **Q2.**  Why strong augmentation is needed? Why not we just use vanilla augmentation from the first time to train the adversarial self-supervised learning model where the model will always have a small training-test gap? I am just curious about the authors’ thoughts.
> > >
> > > **A2.** This is because the drive engine of self-AT, contrastive learning, requires stronger augmentations as a ***necessary*** component to learn good representations in the first place. In the Introduction (3nd paragraph, quoted below), we explain this property from both theoretical and empirical aspects:
> > > > contrastive learning **relies crucially on strong data augmentations** to obtain generalizable representations. **As noted by recent theoretical understandings of contrastive learning (HaoChen et al., 2021; Wang et al., 2022)**, strong augmentations create support overlap between intra-class samples such that the alignment between augmented pairs could cluster these intra-class samples together. **Under weaker augmentations, the accuracy of standard contrastive learning (SimCLR) will dramatically decrease**, as the red line shows in Figure 1(b).
> > >
> > > Therefore, under weak augmentations, standard contrastive learning cannot attain good accuracy in the first place, and consequently, adversarial contrastive learning will also have limited robustness. The study of augmentation strength in Sec 3.2 further verifies this point. As shown in Figure 4c. under very weak augmentations,  e.g., s=0 or 0.1, **both accuracy and robustness of ACL drop dramatically**.
> > >
> > > ---
> > > **Q3.** As I know, SimSiam does not use contrastive learning loss to train the model. How did you apply DynACL on top of SimSiam? And is the ACL in the A5. Table b typo of SimSiam?
> > >
> > > **A3.** Indeed the ACL in A5 (table b) is a typo here and it should be SimSiam. We have fixed it now. SimSiam is a **non-contrastive** method and it does not require negative samples as in SimCLR. Its learning objective is solely a cosine similarity loss between positive pairs, and it utilizes a predictor and stop gradient to avoid feature collapse.
> > >
> > > We adapt it to the existing ACL framework by 1) replacing the (both attack and learning) loss function from the InfoNCE loss to the SimSiam similarity loss; and 2) adding the predictor and stop gradient mechanisms in SimSiam. All the other configurations are kept the same as ACL. When applying DynACL++ to this SimSiam variant of ACL, we also modify it in the same way as to ACL by adopting a dynamic augmentation schedule and a dynamic version of the SimSiam loss (Sec 4.1). We have added a formal description of this variant in **Appendix D.4**. Please refer to there for more details.
> > >
> > > ---
> > > Thanks again and hope our explanations above could address your concerns. We are happy to address your further question and please let us if there is more to clarify.

---

> ### Author Response · Authors · 2022-11-13
> **Response to Reviewer umnn (1/3)**
>
> We express our sincere gratitude to Reviewer umnn for appreciating the novelty and effectiveness of our work. We address your main concerns below.
>
> ---
>
> **Q1.** The difference between the DynACL++ and AdvCL post-processing which pseudo labels the unlabeled examples with k mean clustering.
>
> **A1.** **First**, we want to emphasize that **the key design of our method is the dynamic training strategy** in DynACL, and **DynACL alone (without post-processing) has attained state-of-the-art robustness among self-AT methods** (Table 1). The post-processing in DynACL++ mainly serves as a bonus part instead of the core mechanism. **Second**, pseudo labeling with k-means is a classical idea widely exploited in self-supervised learning, such as DeepClustering [1] and PCL [2]. Moreover, AdvCL and DynACL++ explore **two different ways of applying k-means to self-AT**.
>
> **Key Differences**. **AdvCL applies k-means to pretraining**, which has two drawbacks: 1) As SSL learns slowly, features at earlier training stages are not good enough. Therefore, the pseudo labels that have some mislabeling errors may bias data representations by a large degree. 2) The additional clustering stage also brings **significant computation overhead**, making AdvCL **twice slower than ACL**.
> Given these limitations, we proposed two effective designs in DynACL++: 1) **apply clustering only on pretrained features**, from which we can obtain good pseudo labels and save much computation; 2) to better adapt pretrained features to the downstream classifier, we propose **a new training strategy: linear probing then adversarially full finetuning (LP-AFF)**, which aligns the linear head with the encoder before pseudo AT. Both techniques are shown to be very effective in practice.
>
> **Comparison.** As shown below (quoted from Tab 2&4), when both are using CIFAR-10 pseudo labels, **AdvCL has no actual improvement over ACL, while our post-processing method improves ACL by a large margin (5.6% AA robustness)**. Meanwhile, ACL + post-processing only takes **1/3 training time** of AdvCL. Therefore, our post-processing method is a **much more efficient and effective strategy of applying k-means to self-AT**. We believe this message is new and worthy of note to the community for developing better self-AT methods. We have elaborated their differences better in the revision (**Section 4.2**).
>
> | | AA(%) | SA(%) | Training Time |
> | - | - | - | -|
> | ACL (baseline) | 37.62 | 79.32 | 32.7h |
> | ACL + kmeans in pretraining (AdvCL) |  37.46 | 73.23 | 105h|
> | **ACL + kmeans in post-processing (ours)**  | **43.24**  | **77.40** | **33.6h**|
>
> [1] Deep Clustering for Unsupervised Learning of Visual Features. ECCV 2018. https://openaccess.thecvf.com/content_ECCV_2018/papers/Mathilde_Caron_Deep_Clustering_for_ECCV_2018_paper.pdf
>
> [2] Prototypical contrastive learning of unsupervised representations. ICLR 2021. https://arxiv.org/pdf/2005.04966.pdf
>
> [3] Virtual adversarial training: a regularization method for supervised and semi-supervised learning. TPAMI. 2018. https://arxiv.org/pdf/1704.03976.pdf
>
> [4] When Does Contrastive Learning Preserve Adversarial Robustness from Pretraining to Finetuning?
> https://arxiv.org/pdf/2111.01124.pdf
>
> ---
> **Q2.** Does proposed technique also can be adapted to RoCL? And also boost the performance of this method too?
>
> **A2.** Yes, because we mainly aim to improve the data augmentation strategy widely adopted in self-AT, including RoCL. The performance on CIFAR-10 & STL-10 is shown below:
> #### *1) CIFAR-10 with RoCL backbone*
> | | AA(%) | SA(%) |
> | - | - | - |
> | RoCL  | 26.12 | 77.90 |
> | DynACL++  (ours) | **39.72** | 74.84 |
>
> #### *2) STL-10 with RoCL backbone*
> | | AA(%) | SA(%) |
> | - | - | - |
> | RoCL | 26.51 | 78.19 |
> | DynACL++ (ours) | **41.61** | 71.51 |
>
> It shows that RoCL incorporated with DynACL++ significantly improves over RoCL (13.6% on CIFAR-10 and 15.1% on STL-10 in AA robustness). This experiment demonstrates the broad applicability of our method.
>
> ---
> **Q3.** Does dynamic scheduling also benefit the performance of vanilla SimCLR?
>
> **A3.** Following your advice, we also evaluate vanilla SimCLR with different augmentation strengths and our dynamic schedule. The results are shown below.
> #### *SimCLR with different augmentation strengths  on CIFAR-10*
> | aug strength $s$ | 0 | 0.2 | 0.4 | 0.6 | 0.8 | 1.0 | dynamic($1\rightarrow0$) |
> | - | - | - | - | - | - | - | - |
> | SA(%) | 33.07 | 69.86 | 79.34 | 84.53 | 87.99 | 89.82 | 85.92
>
> (more below)

---

### Author Response · Authors · 2022-11-13
**A Summary of Paper Updates**

We thank all reviewers for their constructive comments. We have updated the paper accordingly with the following major changes:
- **Section 4.2.** Add detailed comparison between our DynACL(++) and other self-AT methods.

- **Appendix D.3.** Add more experimental studies on the augmentation strength, including comparisons among augmentation strategies and the effect of augmentation strength in self-supervised standard training.

- **Appendix D.4.** Add extensive experiments on the generalizability of DynACL++, where it shows **consistent and significant improvements** when generalized to different self-AT methods (RoCL), different SSL frameworks (SimSiam), and different backbone networks (AlexNet).

- **Appendix E.2**: Add **theoretical justification** for the effect of augmentation strength on the classwise distance.

---

### Public Comment · ~Chaoning_Zhang1 · 2023-02-27
**A highly related paper "Decoupled Adversarial Contrastive Learning for Self-supervised Adversarial Robustness" is missing from the reference**

Dear authors,

We are the authors of  the work titled "Decoupled Adversarial Contrastive Learning
for Self-supervised Adversarial Robustness", which was published at ECCV2022.

Our work also investigated adversarial contrastive learning.
It was online two months before the submission of ICLR2023. It would be appreciated that our work can discussed and compared in your camera-ready version.

Thanks for your consideration.

---

> ### Author Response · Authors · 2023-03-02
> **Response**
>
> Dear Chaoning,
>
> Thanks for pointing out your paper! We have added discussion in the Related Works section. However, we are sorry we cannot compare with your results for now since there are some bugs in your released code. If the bugs are fixed, we are happy to do the comparison in the following version. Moreover, according to the ICLR authors' guidelines, we are not supposed to compare with papers published within 4 months before the submission deadline, while your paper was available on arXiv until Jul. 2022, less than 2 months before the deadline.
>
> Best Regards,
>
> Authors

---

### Decision · Program_Chairs · 2023-01-20

**Decision:**

Accept: poster

**Justification For Why Not Higher Score:**

The authors only evaluate the proposed algorithm on smaller datasets, but either contrastive learning or adversarial robustness have already nowadays been evaluated on larger dataset imagenet, or at least tiny imagenet dataset. Also, though the authors provided some theoretical analyses, its connection with the algorithm is quite weak.

**Justification For Why Not Lower Score:**

 The authors showed extensive experiments with surprisingly good performance to support their observation on the data argumentation within self-supervised adversarial training. Though the proposed algorithm could be heuristic and its connection with the theory could be a bit weak, the empirical findings in this paper could be inspiring.

**Metareview: Summary, Strengths And Weaknesses:**

This paper studied the strength of data augmentation in self-supervised adversarial training (self-AT). In particular, the authors proposed a simple remedy named DynACL (Dynamic Adversarial Contrastive Learning) by adopting a dynamic augmentation schedule along the training process. Besides the empirical results about the impact of data augmentation on self-supervised adversarial training, the authors also tried to include some theoretical analyses.

**Note From Pc:**

if the above contains the word "oral" or "spotlight" please see: "oral" presentation means -> notable-top-5% and "spotlight" means -> notable-top-25%. As stated in our emails, we are disassociating presentation type from AC recommendations